# Macrogenomic and Metabolomic Analyses Reveal Mechanisms of Gut Microbiota and Microbial Metabolites in Diarrhea of Weaned Piglets

**DOI:** 10.3390/ani14162327

**Published:** 2024-08-12

**Authors:** Fei Xie, Mei Zhou, Xiaojin Li, Shenghe Li, Man Ren, Chonglong Wang

**Affiliations:** 1College of Animal Science, Anhui Science and Technology University, Chuzhou 239000, China; xiefei19980430@126.com (F.X.); lixj@ahstu.edu.cn (X.L.); lish@ahstu.edu.cn (S.L.); 2Anhui Province Key Laboratory of Animal Nutritional Regulation and Health, Chuzhou 233100, China; 3Institute of Animal Husbandry and Veterinary Medicine, Anhui Academy of Agricultural Sciences, Hefei 230031, China; 1229zhoumei@163.com

**Keywords:** weaned piglets, diarrhea, gut microbiota, macrogenomic, metabolomics

## Abstract

**Simple Summary:**

Recent studies have shown a correlation between piglet diarrhea and the gut microbiota. However, the precise mechanism by which intestinal microorganisms and their metabolites influence diarrhea in weaned piglets remains unclear. This study explored differences in the gut microbiota and associated metabolites between healthy and diarrheic-weaned piglets using macrogenomic and metabolomic analyses. We identified microorganisms that differed markedly between the two groups. We noticed that the number and abundance of *Bacteroidaceae bacterium* were strongly negatively correlated with diarrhea in weaned piglets, whereas *Caudoviricetes* exhibited a strong positive correlation. Our screening for markedly different metabolites such as carnosine, pantothenic acid, pyridoxal, methylimidazoleacetic acid, indole-3-acetaldehyde and 5-hydroxyindoleacetic acid, along with the identification of metabolite-related metabolic pathways like tryptophan and vitamin B_6_ metabolism, identified important pathways associated with gut microbiota dysbiosis caused by post-weaning diarrhea in piglets. Conjoint analysis revealed that a substantial increase in the number and abundance of *Caudoviricetes*, coupled with a decrease in the number and abundance of *Bacteroidaceae bacterium* could potentially serve as microbial markers of post-weaning diarrhea.

**Abstract:**

Recent studies have shown a correlation between piglet diarrhea and the gut microbiota. However, the precise mechanism by which intestinal microorganisms and their metabolites influence diarrhea in weaned piglets remains unclear. This study explored differences in the gut microbiota and associated metabolites between healthy and diarrheic-weaned piglets using macrogenomic and metabolomic analyses. The histomorphological results showed that diarrheic piglets had shorter jejunal and ileal villi, some of which were shed, compared to healthy piglets. Substantial differences in gut microbial diversity and metabolites were also observed, with *Bacteroidaceae bacterium* and *Caudoviricetes* being the main differential organisms that were strongly correlated with host status. Microbial functions, mainly the metabolism of carbohydrates, glycans, lipids, and amino acids, as well as related enzyme activities, were substantially different. The major differential metabolites were carnosine, pantothenic acid (vitamin B_5_), pyridoxal, methylimidazoleacetic acid, indole-3-acetaldehyde, and 5-hydroxyindoleacetic acid. These metabolites were enriched in beta-alanine, histidine, tryptophan, and vitamin B_6_ metabolism, and in the pantothenate and CoA biosynthesis pathways. Combined macrogenomic and metabolomic analyses revealed that carnosine, vitamin B_5_, and pyridoxal were negatively correlated with *Caudoviricetes*; methylimidazoleacetic acid, indole-3-acetaldehyde, and 5-hydroxyindoleacetic acid were positively correlated with *Caudoviricetes*. Whereas carnosine and vitamin B5 were positively correlated with *Bacteroidaceae bacterium*, 5-hydroxyindoleacetic acid was negatively correlated. The decreased abundance of *Bacteroidaceae bacterium* and the increased abundance of *Caudoviricetes* and related metabolites likely contribute to post-weaning diarrhea in piglets. Therefore, the abundance of *Bacteroidaceae bacterium* and *Caudoviricetes* can likely serve as potential markers for identifying and preventing diarrhea in post-weaning piglets.

## 1. Introduction

Piglet diarrhea is a relatively common problem in pig farming, especially in intensive systems. To improve production efficiency, piglets are typically weaned within 3–4 weeks after birth [1]. However, environmental factors such as separation from the sow, changes in the rearing environment, and dietary adjustments can cause diarrhea in weaned piglets [2]. This condition, if left unattended, can lead to death [3], causing huge economic losses to the global pig farming industry [4].

The digestive system of weaned piglets is underdeveloped, and the intestinal mucosal barrier is susceptible to damage [5]. This vulnerability means that pathogenic bacteria can easily colonize the intestinal tract. Upon invasion, these pathogenic bacteria make use of the undigested proteins and other nutrients in the intestines of weaned piglets for their own reproduction and growth. Consequently, intestinal homeostasis is disrupted, affecting the ratio of beneficial and harmful bacteria, resulting in intestinal microbial dysbiosis.

In the gastrointestinal tract of mammals, including pigs, the large number and types of microorganisms, mainly bacteria, viruses, and fungi, are known as the “microbiota”. The microbiota can regulate the metabolic state of the body and the homeostasis of the intestinal immune system. By doing so, the microbiota fortifies the body’s defense mechanism against the invasion of pathogenic bacteria [6,7]. Numerous studies have shown that dysbiosis of the gut microbiota is one of the main causes of diarrhea in weaned piglets [8,9]. Therefore, studying the gut microbiota of weaned piglets may be useful for the prevention and diagnosis of early diarrhea [10,11].

There is a symbiotic relationship between the gut microbiota and the host, characterized by interdependence and mutual constraint [12]. In this relationship, the host provides nutrition and a conducive environment for the gut microbes, while the gut microbes, in turn, regulate the host’s immune system [13]. This results in maintaining the host’s intestinal barrier function and shielding the host from invasion by pathogenic bacteria. Furthermore, interactions between gut microbes and the host produce a large number of metabolites, which shape the basic survival environment of the mammalian gastrointestinal tract and affect host health [14]. In recent years, the correlation between gut microbes, gut microbial metabolites, and host health has attracted increasing attention [15,16]. However, the mechanism underlying the role of gut microbes and their metabolites in diarrhea in weaned piglets remains unclear.

Therefore, to explore whether changes in gut microbes and their metabolites are correlated with post-weaning diarrhea in piglets, we explored the differences in gut microbes and their associated metabolites between weaned healthy and diarrheic piglets based on macrogenomic and metabolomic analyses. In addition, we examined the relationship between gut microbes and their metabolites in weaned piglets experiencing diarrhea by combining macrogenomic and metabolomic analyses. The aim was to identify key microbes implicated in the onset of diarrhea. The results of this study provide a novel theoretical basis for the development of new methods to prevent and control diarrhea in weaned piglets. Moreover, our results may also provide new insights into the screening of microbiota relevant to human intestinal health.

## 2. Materials and Methods

### 2.1. Ethics Statement

Experiments were performed according to the Regulations for the Administration of Affairs Concerning Experimental Animals and approved by the Animal Research Committee of Anhui Science and Technology University (license number 2023-017).

### 2.2. Animals

The piglets used in this study were obtained from Huangshan Guoda Eco-Agriculture Science and Technology Co., Ltd. (Yixian, Huangshan, Anhui Province, China) and were fed and maintained in the same environment in accordance with standard feeding management practices. None of the piglets were treated with antibiotics. The farm had no history of *porcine reproductive* and *respiratory syndrome virus*, porcine circovirus type 2, or *porcine epidemic diarrhea virus* infections. All piglets were weaned at 30 d of age. The criterion for determining whether a piglet had diarrhea was that their feces were watery or liquid for at least 5 d and solid for 5 d in healthy piglets. Finally, eight weaned piglets (*n* = 8; 37 ± 3 d of age; multiple litters; siblings or half-siblings; half male and half female) were selected for this experiment, of which four were diarrheal piglets and four were healthy piglets that had similar dates of birth and litter sizes.

### 2.3. Sample Collection

Blood was collected from the anterior vena cava of piglets, and plasma samples were collected by centrifugation (5000× *g*, 10 min) and stored at −80 °C. After slaughter, rectal fecal samples were collected from each piglet and immediately frozen in liquid nitrogen. Segments of the jejunal and ileal intestines were collected, rinsed with phosphate-buffered saline, fixed (4% paraformaldehyde solution), and used for subsequent morphological observations.

### 2.4. Morphological Structure of the Intestinal Tract

The fixed jejunal and ileal intestinal segments were removed, and approximately 1 cm-long intestinal segments were collected, dehydrated, rendered transparent, and embedded in wax blocks. Then the wax blocks were sectioned into 5 µm-thick slices, stained using the hematoxylin–eosin (HE) staining protocol [17], and the morphological structure of the intestinal tract was observed using a light microscope with a pathology section observation system.

### 2.5. Macrogenome and LC–MS Untargeted Metabolomics Sequencing and Analysis

The frozen fecal and blood samples were sent to Shanghai Meiji Biomedical Technology Co., Ltd. (Shanghai, China) for macrogenomic and metabolomics sequencing, as described in the online Appendix A.

### 2.6. Combined Analyses of Macrogenomic and Metabolomic Processes

To investigate the correlation between gut microbial species and metabolites, differential microbes screened by macrogenomes and differential metabolites screened by metabolomes were subjected to Spearman’s correlation analysis. Correlation analyses were performed using SPSS version 26 (IBM, New York, NY, USA) and Origin software version 2021 (IBM, New York, NY, USA). The values of |r| > 0.6 and *p* < 0.05 were considered to indicate strong correlations.

## 3. Results

### 3.1. Morphological Structure of the Intestine

In the present study, the intestinal mucosal tissues of the jejunum and ileum of healthy piglets and diarrheic piglets were morphologically compared using HE staining (Figure 1). The jejunum and ileum of healthy piglets (Figure 1A,C) were compared with those of diarrheic piglets (Figure 1B,D). The results showed that the length of the intestinal villi became shorter in the latter group and some also underwent detachment.

### 3.2. Quality Assessment of Macrogenomic Sequencing Data

The macrogenomic sequencing results of the gut microbial samples from healthy and weaned piglets with diarrhea are shown in Table 1. The total sequencing volume of the samples was 37,836.37 Mb, and the average data volume from each piglet was 4729.55 Mb. After the quality control process, the total valid data and valid average data volume per piglet were 36,914.35 Mb and 4614.29 Mb, respectively. The effective rate of quality control was 97.6%, indicating that the samples in this experiment were of high quality and met the requirements for assessing sequencing depth and subsequent tests.

### 3.3. Gut Microbial Composition

No difference (*p* > 0.05) was observed between healthy and diarrheic piglets in the Chao1, Shannon, and Simpson indices for gut microbial alpha diversity (Figure 2). The gut microbial beta diversities of healthy and diarrheic piglets were assessed using principal coordinate analysis (PCoA). The gut microbes of healthy piglets were differentiable from those of diarrheic piglets (Figure 2D). Beta diversity was verified using the Wilcoxon rank–sum test and the gut microbial structures of healthy piglets and diarrheic piglets were different (*p* < 0.05; Figure 2E).

At the phylum level, *Firmicutes*, *Bacteroidota*, and *Proteobacteria* were the major taxa in healthy and diarrheic piglets, and their relative abundance was more than 0.5% of all bacteria (Figure 3A). *Firmicutes* was the predominant phylum, accounting for more than 68.71% and 56.89% of the total bacterial load in diarrheic and healthy piglets, respectively. At the genus level, the 10 genera with high relative abundance in the gut microbiota of healthy piglets were *Lactobacillus*, *unclassified_o_Bacteroidales*, *unclassified_o_Eubacteriales*, *Prevotella*, *Bacteroides*, *unclassified_f_Oscillospiraceae*, *unclassified_f_Lachnospiraceae*, *unclassified_p_Firmicutes*, *unclassified_f_ Muribaculaceae*, and *Streptococcus*, and the 10 genera with high relative abundance in the gut microbiota of diarrheic piglets were *Lactobacillus*, *unclassified_f_Lachnospiraceae*, *Clostridium*, *unclassified_o_Eubacteriales*, *Klebsiella*, *Mitsuokella*, *unclassified_f_Enterobacteriaceae*, *Prevotella*, *Megasphaera*, and *unclassified_c_Clostridia* (Figure 3B). The 20 species with the highest relative abundance in the gut microbes of healthy and diarrheic piglets are shown in Figure 3C.

Some significant changes in the microorganisms at the phylum, order, and species levels were observed in healthy and diarrheic piglets. At the phylum level, the relative abundance of *Bacteroidota* was substantially higher in healthy piglets than in diarrheic piglets, and the relative abundance of *Uroviricota* was substantially lower than that in weaned diarrheal piglets (Figure 4A). At the order level, the relative abundance of *Bacteroidia* was substantially higher in healthy piglets than that in diarrheic piglets, and the relative abundance of *Caudoviricetes* was substantially lower in healthy piglets than that in diarrheal piglets (Figure 4B). At the species level, the relative abundances of *Siphoviridae_*sp., *Podoviridae_*sp. and *CrAss-like_virus_*sp. were substantially higher in weaned diarrheal piglets than in healthy piglets.

Statistical differences in gut microbial composition between healthy and diarrheic piglets were further analyzed using linear discriminant analysis effect size (LefSE) (Figure 5A). The gut microbes of healthy and diarrheic piglets varied in the core group of bacteria from the phylum to the order of *Bacteroidetes*. Some strains belonging to the phylum *Bacteroidetes*, the class *Bacteroidia*, and the order *Bacteroidales* scored higher in the healthy, weaned piglets (Figure 5B). The viruses were also substantially different between healthy and diarrheic piglets (Figure 5C), with the phylum *Uroviricota* and the class *Caudoviricetes* scoring higher in diarrheal piglets. Higher linear discriminant analysis (LDA) scores indicated that the strain contributed to diarrhea to a greater extent (LDA > 3.5; *p* < 0.05) (Figure 5D).

### 3.4. Correlation between the Number and Abundance of Gut Microbiota and Diarrhea in Weaned Piglets

A strong negative correlation existed between the abundance of *Bacteroidaceae bacterium* and diarrhea in the weaned piglets (*p* < 0.05; r < −0.6), and strong positive correlations between the abundances of *Caudoviricetes*, *Siphoviridae_*sp., *Podoviridae_*sp. and *CrAss-like_virus_*sp. were strongly positively correlated with diarrhea in the weaned piglets (*p* < 0.05; r > 0.6) (Figure 6).

### 3.5. Functional Characteristics of the Gut Microbiota

In this study, 4,443,593 open reading frames (ORFs) were detected by metagenome sequencing, with an average of 555,449 ORFs per sample. A total of 3,782,670 ORFs were obtained following quality control inspection for subsequent functional analyses. The total length of the non-redundant ORFs was 2077.80 Mb, with an average length of 259.73 bp per sample. KEGG functional analysis results showed that the microbial functions of healthy and diarrheic piglets were concentrated in metabolism, genetic information processing, environmental information processing, cellular processes, human diseases, and organismal systems. Carbohydrate, amino acid, cofactor and vitamin, energy and nucleotide metabolism, as well as glycan biosynthesis and metabolism, were enriched in healthy and diarrheic piglets (Figure 7A). No obvious clustering of KEGG functions was observed in the samples of healthy and diarrheic piglets (Figure 7B).

In the present study, Welch’s test was used to identify statistical differences in microbial functions between healthy and diarrheic piglets. Carbohydrate, lipid, and amino acid metabolism and glycan biosynthesis and metabolism were the main factors involved (Figure 8). Compared with the healthy piglets, carbohydrate metabolism (citrate and tricarboxylic acid cycles; ko00020; *p* < 0.05), glycan biosynthesis and metabolism (other glycan degradation; ko00511; *p* < 0.05), glycosphingolipid biosynthesis (globo and isoglobo series; ko00603; *p* < 0.05), N-glycan biosynthesis (ko00510; *p* < 0.05), and lipid metabolism (sphingolipid metabolism; ko00600; *p* < 0.05) were down-regulated in piglets with diarrhea. However, amino acid metabolism (tyrosine degradation; ko00350; *p* < 0.05) was up-regulated in piglets with diarrhea.

### 3.6. Metabolomics Analysis

In this study, LC–MS-based untargeted metabolomics was used to identify markedly altered metabolites in the blood of healthy and diarrheic piglets, Partial least squares-discriminant analysis (PLS-DA) and orthogonal partial least squares-discriminant analysis (OPLS-DA) were used to analyze the differences in metabolites between the two groups (Figure 9). Different metabolic characteristics were observed in the blood samples of healthy and diarrheic piglets. The PLS-DA and OPLS-DA models were validated in the present study, and the models of PLS-DA and OPLS-DA in the positive and negative ion modes were stable and reliable (Figure 10). As indicated by the volcano plots (Figure 9E,F), a total of 91 and 93 metabolites were changed in the positive ion mode (48 up and 43 down; Figure 9E) and negative ion mode (53 up and 40 down; Figure 9F) of the weaned, diarrheic piglets. These results indicate that the metabolites of piglets with diarrhea change substantially. Using MS–MS analysis, the peak characteristics and identification of metabolites changed markedly. The results of the appraisal analysis indicated that most of the important changes in metabolic products from amino acid synthesis and their derivatives, organic acids and indole derivatives, lipids, fatty acids, and their metabolites were directly produced by microorganisms or modified by them. MetaboAnalyst enrichment analysis was used to determine the differentially expressed metabolites in biological pathways and their biological effects. The significant metabolic pathways (*p* < 0.05) mainly included beta-alanine, histidine, tryptophan, and vitamin B_6_ metabolism in addition to pantothenate and CoA biosynthesis (Figure 11).

### 3.7. Combined Analysis of the Gut Microbiome and Blood Metabolome

To explore the correlation between gut microorganisms and their metabolites, this study conducted a joint analysis of changed gut microorganisms and metabolites based on the results of macrogenomics and metabolomics and screened the top 35 changed metabolites by using the screening criteria of VIP > 1 and *p* < 0.05 in the OPLS-DA model (Appendix A). The top 10 changed metabolites were screened using the LDA > 3.5 and *p* < 0.05 screening criteria. The top 10 changes in the relative abundance of gut microorganisms were screened for analysis using Spearman correlations. A correlation was found between changed microorganisms and changed metabolites (*p* < 0.05, |r| > 0.7). Of these, carnosine, pantothenic acid, and pyridoxal showed negative correlations with *Caudoviricetes.* Methylimidazoleacetic acid, indole-3-acetaldehyde, and 5-hydroxyindoleacetic acid were positively correlated with *Caudoviricetes*. Carnosine and pantothenic acid were positively correlated with *Bacteroidaceae bacterium*, and 5-hydroxyindoleacetic acid was negatively correlated with *Bacteroidaceae bacterium* (Figure 12).

## 4. Discussion

Gut microbial dysbiosis is one of the main factors causing diarrhea in weaned piglets and is also a leading cause of death in piglets. Although some studies have reported that gut microbial dysbiosis is closely related to diarrhea in weaned piglets, the mechanisms involving the microorganisms and their metabolites remain unclear.

When diarrhea strikes weaned piglets, it triggers changes in their intestinal morphology, physiological functions, and gut microbiota. Although the small intestine primarily serves as the main site for nutrient digestion and absorption, it also acts as a crucial barrier against pathogens and physiological stress [18]. Structurally, the small intestine is composed of the mucosa, submucosa, and muscularis propria and the intestinal villi are specialized structures in the mucosal layer. The integrity and oscillation of these villi can effectively inhibit the growth of pathogenic microorganisms colonizing the intestine [19]. During diarrhea episodes in piglets, the intestinal tissues suffer damage; the intestinal villi are atrophied and detach at different levels. In severe cases, inflammatory cell infiltration may also occur [20]. Our study showed a decrease in the length of the jejunum and ileum villi of weaned piglets with diarrhea and some of the villi were detached. This observation is consistent with a study by Zheng et al. [21]. Collectively, these observations suggest that diarrhea in weaned piglets may change the morphological structure of their intestinal tissues.

In the intestinal tract of mammals, there is a symbiotic microbial system known as the intestinal microbiome, which is composed of bacteria, fungi, and viruses, amongst others. This microbiome plays a crucial role in defense against pathogens, promoting immune system development, and maintaining intestinal barrier function [22,23]. Changes in the composition and function of gut microbes, termed microbial dysbiosis, are important factors affecting various inflammatory bowel diseases, such as inflammatory bowel disease (IBD) and diarrhea. In the present study, we observed marked differences in gut microbial composition and function between weaned piglets and those with diarrhea. Among the main differential microorganisms identified were *Bacteroidaceae bacterium* and *Caudovirales*.

*Bacteroidaceae bacterium* consists of *Bacteroidota* and *Bacteroidetes*, which are stable commensal bacteria in the mammalian gastrointestinal tract. These bacteria are known for their ability to hydrolyze proteins and produce butyrate. By breaking down complex polysaccharides, proteins, and lipids, *Bacteroides* enhance intestinal digestion and absorption, thereby supporting the growth of other commensal bacteria in the gut and maintaining intestinal homeostasis [24]. Our study found marked differences between the healthy piglets and those experiencing diarrhea. Carbohydrate, lipid, and amino acid metabolism, as well as glycan biosynthesis and metabolism and the activities of some associated enzymes were down-regulated (*p* < 0.05) in diarrheic piglets compared with their healthy counterparts.

Phages present in the host gut are mainly classified into dsDNA *Caudovirales* and non-tailed ssDNA *Microviridae* [25,26]. Together with the rest of the gut microbial community, these phages maintain the dynamic homeostasis of the intestinal tract and are key regulators of intestinal homeostasis and inflammation [27,28]. Although the majority of the enterovirome consists of phages that can infect bacteria [29], in the absence of disease, phage populations in the host gut show substantial diversity and temporary stability [30]. However, dysregulation of the enteroviral group, particularly a substantial increase in *Caudovirales*, was shown to affect intestinal immunity, barrier function, and overall intestinal homeostasis [31]. Studies by Norman et al. [32] and Clooney et al. [33], among others, showed that, compared to healthy populations, patients with Crohn’s disease or ulcerative colitis exhibit a substantially higher abundance of *Caudovirales*. Similarly, a marked increase in *Caudovirales* abundance was observed in both patients with IBD and diarrheic neonatal piglets [34,35]. In the current study, we found that the relative abundance of *Caudovirales* was substantially higher and that of *Bacteroidaceae bacterium* was substantially lower in weaned diarrheic piglets than in healthy piglets. In addition, the number and abundance of *Bacteroidaceae bacterium* were strongly negatively correlated in diarrheic piglets, whereas the number and abundance of *Caudovirales* exhibited a strong positive correlation. Based on these observations, we hypothesized that a substantial decrease in the relative abundance of *Bacteroidaceae bacterium* and an increase in the relative abundance of *Caudovirales* may be one of the main factors contributing to diarrhea in weaned piglets.

The metabolites produced by gut microbes are key mediators of microbe–host interactions. For example, tryptophan, short-chain fatty acids, lipids, and other secondary metabolites play crucial roles in host metabolism, maintaining intestinal homeostasis, and regulating intestinal integrity [36,37,38]. In the present study, we found marked differences between the intestinal microbial metabolites of healthy and diarrheic piglets. These differences were mainly found in amino acids and their derivatives, organic acids and their derivatives, indoles and their derivatives, lipids and lipid-like molecules, and fatty acid concentrations. Of these, indoles and amino acids and their derivatives were shown to be associated with animal intestinal health [39]. Furthermore, the present study found marked differences in the enrichment pathways of intestinal microbial metabolites between healthy and diarrheic piglets. These differences primarily included beta-alanine, histidine, tryptophan metabolism, and vitamin B_6_ metabolism and pantothenate and CoA biosynthesis. Importantly, tryptophan and vitamin B6 metabolism were associated with animal gut health [40,41]. Therefore, these five altered metabolic pathways may play an important role in the onset and progression of diarrhea in weaned piglets.

Gut microbes convert ingested food or host products into metabolites that can directly impact the gut microbiota or host cells. The presence of metabolites largely depends on the metabolic activities of the gut microbes [42,43]. In the current study, we identified a correlation between altered microorganisms and metabolites using combined macrogenomic and metabolomic analyses (*p* < 0.05; |r| > 0.7). Of these correlations, carnosine, pantothenic acid, and pyridoxal were negatively correlated with *Caudoviricetes*, whereas methylimidazoleacetic acid, indole-3-acetaldehyde, and 5-hydroxyindoleacetic acid were positively correlated with *Caudoviricetes*. Carnosine and pantothenic acid were positively correlated with *Bacteroidaceae bacterium*, whereas 5-hydroxyindoleacetic acid exhibited a negative correlation with *Bacteroidaceae bacterium*. These results further highlight the possibility that *Bacteroidaceae bacterium* and *Caudoviricetes* may be used as microbial markers for diarrhea in weaned piglets.

## 5. Conclusions

In the present study, we used macrogenomics and metabolomics to elucidate differences in gut microbial composition, function, and metabolite changes between healthy piglets and those with diarrhea. Microorganisms were identified that differed substantially between the two groups. We noticed that the number and abundance of *Bacteroidaceae bacterium* were strongly negatively correlated with diarrhea in weaned piglets, whereas *Caudoviricetes* exhibited a strong positive correlation. In addition, our screening for substantially different metabolites such as carnosine, pantothenic acid, pyridoxal, methylimidazoleacetic acid, indole-3-acetaldehyde and 5-hydroxyindoleacetic acid, along with the identification of metabolite-related metabolic pathways such as tryptophan and vitamin B_6_ metabolism, identified important pathways associated with gut microbiota dysbiosis caused by post-weaning diarrhea in piglets. Conjoint analysis further revealed that a marked increase in the number and abundance of *Caudoviricetes*, coupled with a marked decrease in the number and abundance of *Bacteroidaceae bacterium*, could potentially serve as microbial markers for the early detection of post-weaning diarrhea. Therefore, the results of our study provide valuable insights for the prevention and control of diarrhea in weaned piglets in later stages.

## Figures and Tables

**Figure 1 animals-14-02327-f001:**
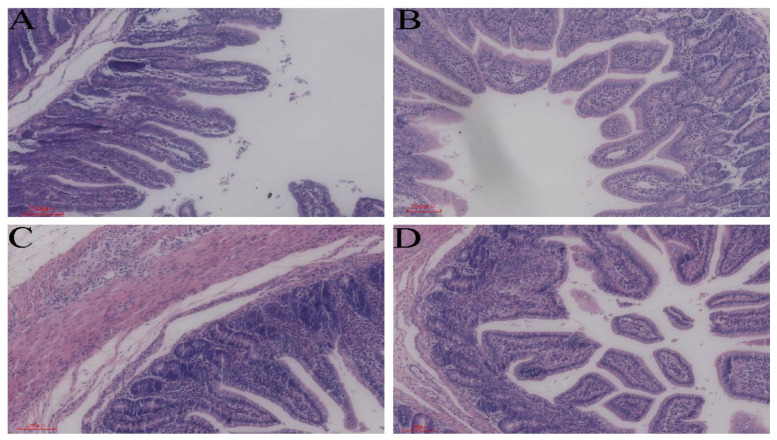
Morphological structure of piglet intestine (100 µm): (**A**) jejunum of weaned healthy piglets; (**B**) jejunum of weaned diarrhoeic piglets; (**C**) ileum of weaned healthy piglets; (**D**) ileum of weaned diarrhoeic piglets.

**Figure 2 animals-14-02327-f002:**
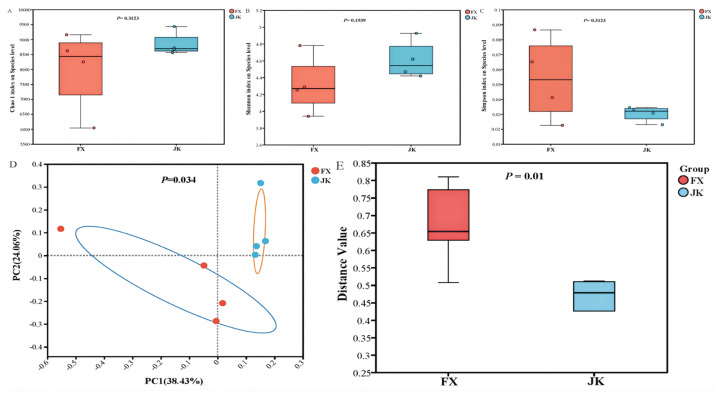
Diversity of gut microbiota between weaned healthy piglets and weaned diarrhoeic piglets. Alpha diversity including Chao1 (**A**), Shannon (**B**) and Simpsons (**C**) was displayed between weaned healthy piglets and weaned diarrhoeic piglets. Beta diversity including the PCoA (**D**) and Wilcoxon rank sum test (**E**) was analyzed between weaned healthy piglets and weaned diarrhoeic piglets.

**Figure 3 animals-14-02327-f003:**
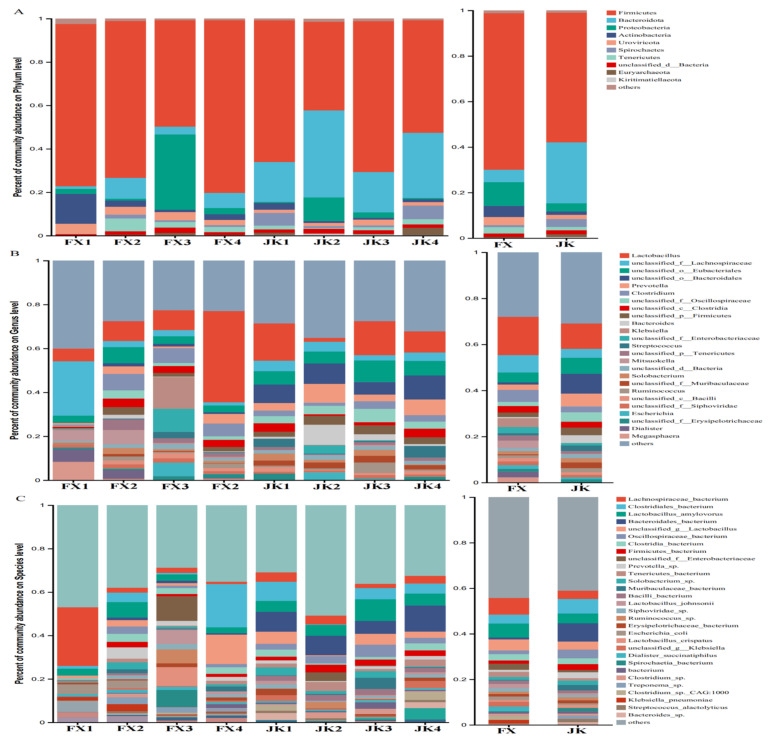
Composition of gut microbiota between weaned healthy piglets and weaned diarrhoeic piglets. Histogram showing top 10 relative abundance of phylum (**A**) and top 25 relative abundance of genus (**B**) and top 35 relative abundance of species (**C**) among samples or groups.

**Figure 4 animals-14-02327-f004:**
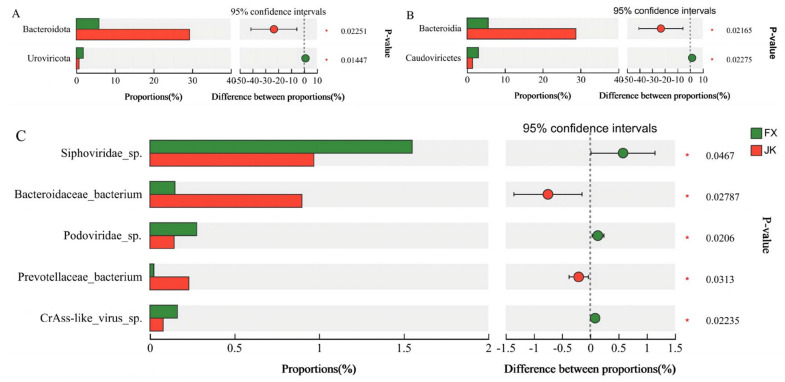
Differential microbiota between weaned healthy piglets and weaned diarrhoeic piglets at the phylum (**A**), class (**B**) and species (**C**) levels. * *p* < 0.05 means significant difference.

**Figure 5 animals-14-02327-f005:**
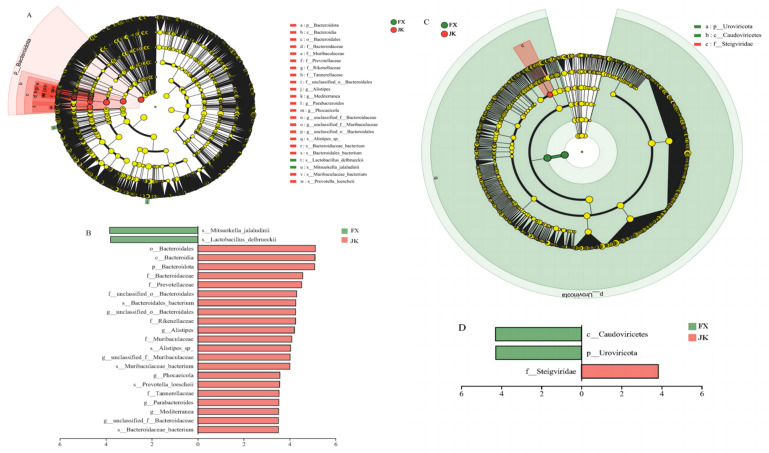
Differences in gut microbiota between weaned healthy piglets and weaned diarrhoeic piglets: (**A**) LefSe bar of the weaned healthy piglets and weaned diarrhoeic piglets from phylum to genus level with cladogram showing; (**B**) LefSe bar of the healthy and diarrhea piglets at species level; (**C**) LefSe bar of the weaned healthy piglets and weaned diarrhoeic piglets from phylum to family level with cladogram showing; (**D**) LefSe bar of the healthy and diarrhea piglets at class and phylum level.

**Figure 6 animals-14-02327-f006:**
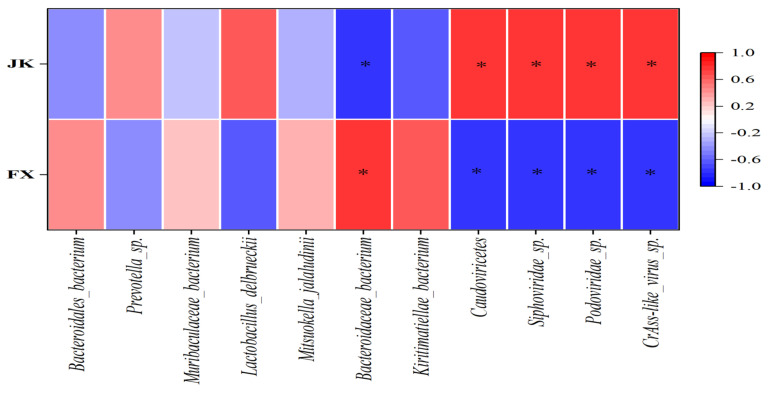
Heatmap of correlation between microbial groups and diarrhea status. * means *p* < 0.05, |r| > 0.6.

**Figure 7 animals-14-02327-f007:**
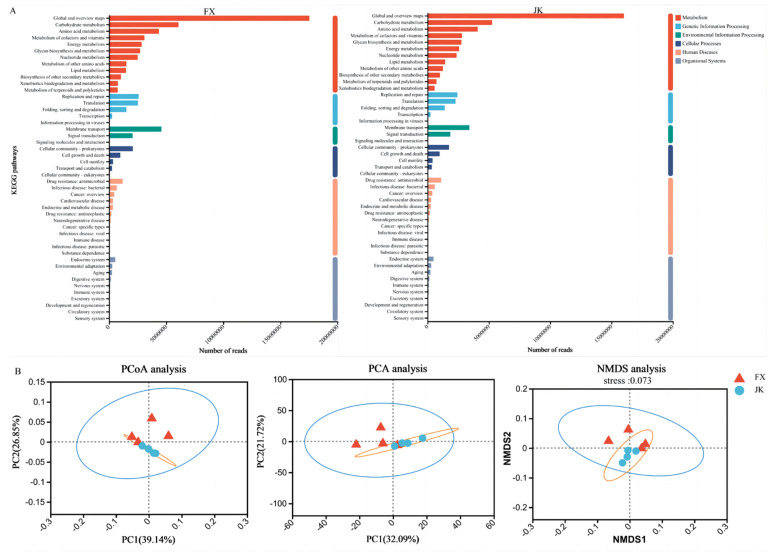
Microbial function characteristics and differences between weaned healthy piglets and weaned diarrhoeic piglets: (**A**) the microbial function characteristics at different levels; the microbial function differences were shown by (**B**) principal component analysis (PCA), principal coordinate analysis (PCoA) and a one-way analysis of similarity (ANOSIM) test.

**Figure 8 animals-14-02327-f008:**
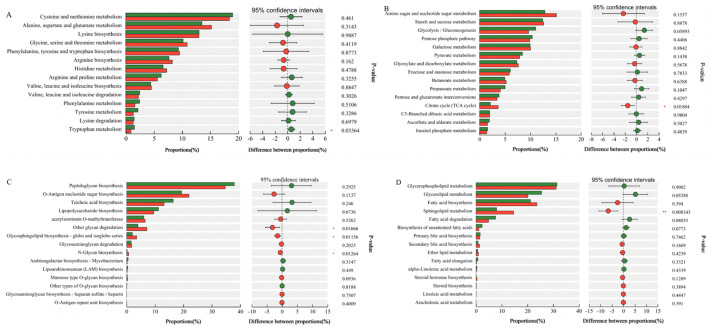
Differences in microbial functions between weaned healthy piglets and weaned diarrhoeic piglets revealed by metagenomic analysis: (**A**) amino acid metabolism; (**B**) carbohydrate metabolism; (**C**) glycan biosynthesis and metabolism; (**D**) lipid metabolism. * *p* < 0.05 means significant difference, ** *p* < 0.01 means extremely significant difference.

**Figure 9 animals-14-02327-f009:**
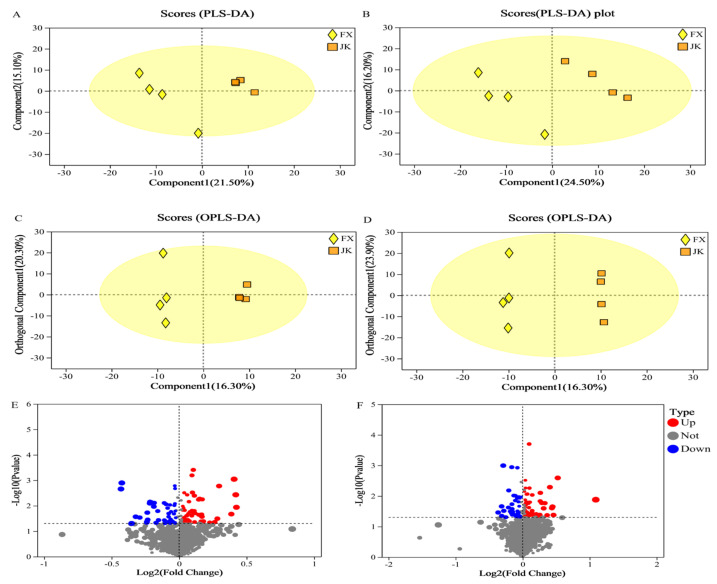
Diarrhea significantly changed the metabolic profiles of weaned healthy and weaned diarrhoeic piglets: (**A**) PLS-DA under positive ion modes; (**B**) PLS-DA under negative ion modes; (**C**) OPLS-DA under positive ion modes; (**D**) OPLS-DA under negative ion modes; (**E**) volcano plot shows the changed metabolites under positive ion modes, weaned diarrhea piglets increased 48 and decreased 43 metabolites when compared with weaned healthy piglets; (**F**) volcano plot shows the changed metabolites under negative ion modes, weaned diarrhea piglets increased 53 and decreased 40 metabolite when compared with weaned healthy piglets.

**Figure 10 animals-14-02327-f010:**
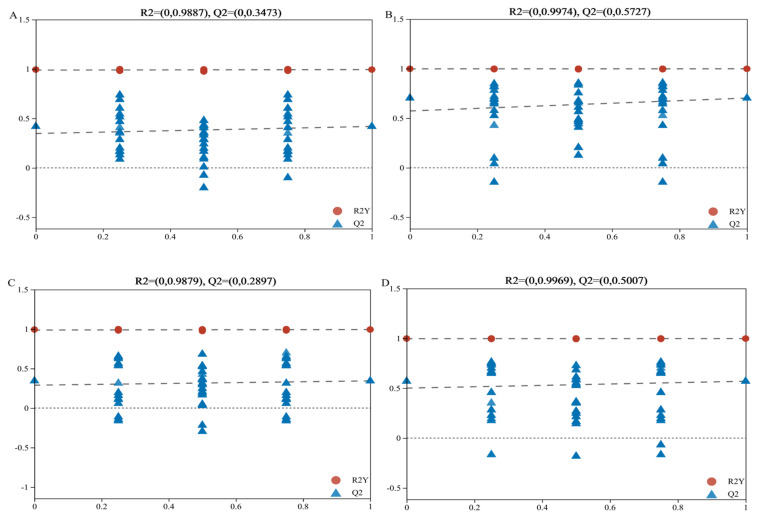
Permutation test of OPLS-DA and PLS-DA model between weaned healthy and weaned diarrhoeic piglets. Red and blue represent weaned healthy and weaned diarrhoeic piglets, respectively. PLS-DA permutation test derived from the blood metabolite profiles of weaned healthy and weaned diarrhoeic piglets at positive (**A**) and negative (**B**) ions mode. OPLS-DA permutation test derived from the fecal metabolite profiles of weaned healthy and weaned diarrhoeic piglets at positive (**C**) and negative (**D**) ions mode.

**Figure 11 animals-14-02327-f011:**
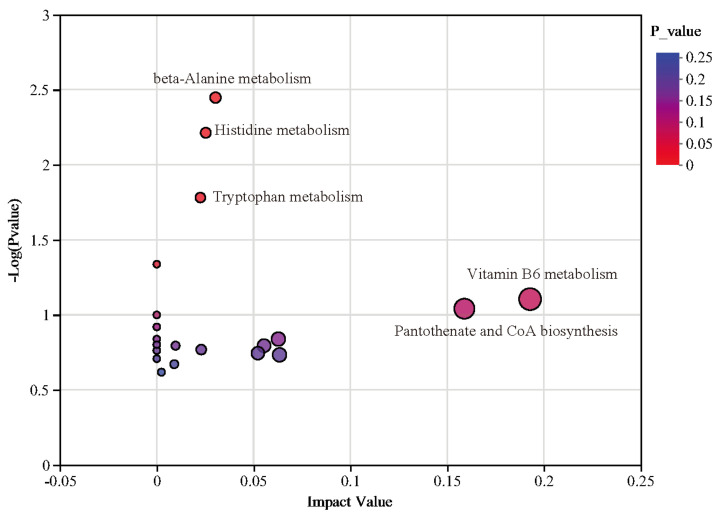
Pathway impact resulting from the differential metabolites using MetaboAnalyst 3.0. Small *p*-value and big pathway impact factor indicate that the pathway is greatly influenced.

**Figure 12 animals-14-02327-f012:**
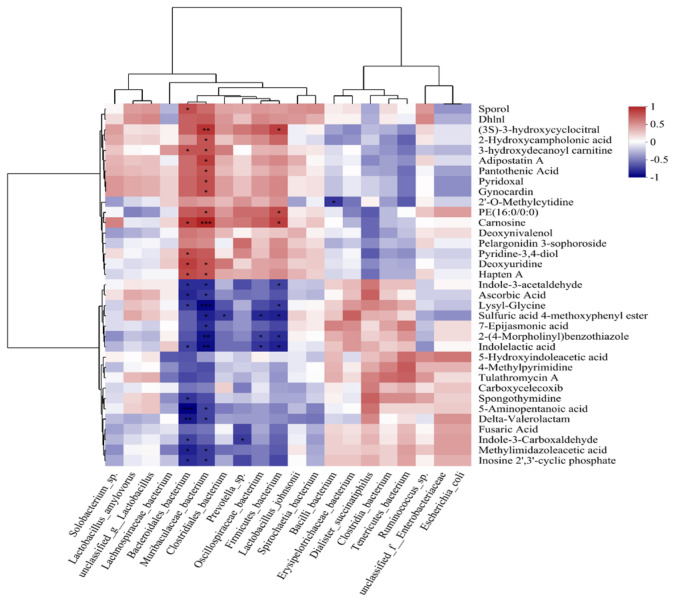
Spearman correlation coefficients between gut metabolites and microbiota. The x-axis shows top 10 bacteria, while the y-axis shows the 35 significantly changed metabolites. * means 0.01 < *p* ≤ 0.05, ** means 0.001 < *p* ≤ 0.01, *** means *p* ≤ 0.01.

**Table 1 animals-14-02327-t001:** Data information of metagenomic illumine sequencing.

Samples ID ^1^	Insert Size (bp)	Read Length (bp)	Raw Reads	Clean Reads	Efficient (%)
FX1	500	150	44,989,508	43,900,536	97.5
FX2	500	150	44,978,778	43,857,974	97.5
FX3	500	150	49,033,976	47,870,534	97.6
FX4	500	150	46,808,674	45,697,624	97.6
JK1	500	150	47,345,902	46,188,976	97.6
JK2	500	150	46,140,104	45,020,146	97.6
JK3	500	150	52,361,694	51,103,206	97.6
JK4	500	150	46,705,024	45,504,642	97.4

^1^ FX = weaned diarrhoeic piglets, JK = weaned healthy piglets.

## Data Availability

The raw data supporting the conclusions of this article will be made available by the authors on request.

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
