# Peer review of "Macrogenomic and Metabolomic Analyses Reveal Mechanisms of Gut Microbiota and Microbial Metabolites in Diarrhea of Weaned Piglets"

_animals, 2024, doi:10.3390/ani14162327_

Round 1

Reviewer 1 Report

Comments and Suggestions for Authors

I am likely not knowledgeable enough on power analyses and their computation in this type of metagenomics and metabolomics and what is a reasonable sample size; however, I would question results being presented for n = 4 of each 'treatment' group (healthy vs. diarrhea) to draw clinically significance from findings. Recognizing that such analyses are very costly (a forever problem in vet med, and not the fault of the authors). Certainly trends pulled from data can direct future studies. Authors should clarify if piglets from multiple litters or from same sow, etc. 

Author Response

The author should clarify whether the piglets are from multiple litters or from the same sow, etc.

Reply:Thank you very much for your suggestion. It has been revised in line 110-111  on newly submitted paper.

Reviewer 2 Report

Comments and Suggestions for Authors

Line 23. Please modify Vitamin B6 to Vitamin B6. Please check the subscripts for vitamins in the manuscript.

Abstract. Please briefly describe this study's materials and methods in the abstract.

Line 59-64, 73-83. Please add the reference.

Line 78. It has been said that many metabolites are produced through interactions between intestinal microorganisms and the host. What are some examples of metabolites and the relationships between intestinal microorganisms and metabolites that have been discovered so far?

Line 102. Were the pigs in a pen during the experiment? Were the pigs in a cage? If it was in a cage, what size would it be?

Line 107. Is there a reference to the criteria for determining whether piglets have diarrhea?

Line 109-111. If you used 4 healthy pigs and 4 pigs with diarrhea, is the number of replicates 4? Doing 4 replicates each is too few replicates.

Line 137. Are there any results showing statistically significant differences from the measured values ​​for villus height, crypt depth, and villus height:crypt depth ratio in the piglet intestinal morphology analysis results?

Line 162-163, 181-182, 193-194, 225-226. Please describe this part in the materials and methods section, not the results.

Line 167-180. Please italicize all names of microorganisms. Please check the entire manuscript.

Line 201. Please italicize the P. Please check the entire manuscript.

Line 273. The full names of PLS-DA and OPLS-DA must be defined on Line 273, not Line 277.

Please double-check that the references are in the format of this journal.

Comments on the Quality of English Language

Please check this manuscript for typos and grammatical errors.

Author Response

Line 23. Modify vitamin B6 to vitamin B6. Please check the subscript for vitamins in the manuscript.

Reply:Thank you very much for your suggestion. It has been revised in the newly submitted paper.

Please provide a brief description of the materials and methods of this study in the abstract.

Reply:Thank you very much for your suggestion. I'm following the journal abstract template

Lines 59-64, 73-83: Please add a reference.

Reply:Thank you very much for your suggestion. It has been revised in the newly submitted paper.

78.It line has already stated that many metabolites are produced through interactions between gut microbes and hosts. What are some examples of metabolites discovered so far and the relationship between gut microbes and metabolites?

Reply:Thank you very much for your suggestion. For instance, both the bile acids and inositol phosphate in the subsequent two articles are generated by microorganisms and can maintain intestinal homeostasis by regulating certain physiological processes within the body.

Wu SE, Hashimoto-Hill S, Woo V, et al. Microbiota-derived metabolite promotes HDAC3 activity in the gut. Nature. 2020; 586(7827):108-112.

Mohanty I, Mannochio-Russo H, Schweer JV, et al. The underappreciated diversity of bile acid modifications. Cell. 2024; 187(7):1801-1818.e20.

Line 102: Were the pigs in the enclosure during the experiment? Are pigs kept in cages? If it's in a cage, how big will it be?

Reply:Thank you very much for your suggestion. All piglets in this trial lived in pens and were 37±3 days old. The weight of weaned healthy piglets was 9-10kg, and the weight of weaned diarrhea piglets was 7-8kg.

Line 107: Did the criteria used to determine whether a piglet have diarrhea were referenced?

Reply:Thank you very much for your suggestion. In this study, the criteria for determining piglet diarrhea have been described in laboratory animals

Lines 109-111: If you use 4 healthy pigs and 4 pigs with diarrhea, is the number of replicates 4? 4 repetitions per repetition is too little.

Reply:Thank you very much for your suggestion. Because pigs are large animals, the normal number of replicates is 3 to 6, and the number of replicates used in this study is 4, which meets the requirements of the number of replicates.

Line 137. In the results of the intestinal morphology analysis of piglets, are there any results that show a statistically significant difference in the measurements of villus height, crypt depth, and villus height:villus depth ratio?

Reply:Thank you very much for your suggestion. There were results to show significant differences in villus height, crypt depth, villus height: crypt depth.

Lines 162-163, 181-182, 193-194, 225-226. Please describe this section, not the results, in the Materials and Methods section.

Reply:Thank you very much for your suggestion. It has been revised in the newly submitted paper.

Lines 167-180: Please italicize all microbial names. Please check the entire manuscript.

Reply:Thank you very much for your suggestion. It has been revised in the newly submitted paper.

Line 201: Italicize P. Please check the entire manuscript.

Reply:Thank you very much for your suggestion. It has been revised in the newly submitted paper.

Line 273. The full names of PLS-DA and OPLS-DA must be defined on line 273 and not on line 277.

Reply:Thank you very much for your suggestion. It has been revised in the newly submitted paper.

Please double-check that the references are in the format of this journal.

Reply:Thank you very much for your suggestion. It has been revised in the newly submitted paper.

Reviewer 3 Report

Comments and Suggestions for Authors

 The manuscript titled “Macrogenomic and metabolomic analyses reveal mechanisms of gut microbiota and microbial metabolites in diarrhea in weaned piglets” is written fluently and there are considerable parameters to justify the achieved results. The title is very relevant and addresses a common issue in weaned pigs. However, there are several errors that need to be addressed. Please check all the references in the manuscript and revise them based on the journal format. For example [22],[23] can be changed to [22,23] etc. moreover, there are too many statements without any citations. For example in lines 53-54, 59-68, etc. Please check the discussion section too. please check all the abbreviations carefully. When you introduce an abbreviation for the first time, spell out the full term and follow it with the abbreviation in parentheses. Once you've defined an abbreviation, use it consistently throughout the manuscript. Don't switch between different abbreviations for the same term. Abbreviations stand alone in the “abstract” and in the main text. For example its unnecessary to abbreviate SCFA in line 394. PRRSV, PCV2, PEDV in line 105-106, PBS in line 117. The figures and Tables should be self-standing and the treatments and abbreviations have to be defined. For example, almost in all figures, please define and explain FX and JK. The figures resolution is low and its unclear why most of the figures are horizontally stretched.

Author Response

Please check all the references in the manuscript and revise them based on the journal format. For example [22],[23] can be changed to [22,23] etc.

Reply:Thank you very much for your suggestion. It has been revised in the newly submitted paper.

moreover, there are too many statements without any citations. For example in lines 53-54, 59-68, etc.

Reply:Thank you very much for your suggestion. It has been revised in the newly submitted paper.

Please check all the abbreviations carefully.

Reply:Thank you very much for your suggestion. It has been revised in the newly submitted paper.

The figures and Tables should be self-standing and the treatments and abbreviations have to be defined.

Reply:Thank you very much for your suggestion. It has been revised in the newly submitted paper.

The figures resolution is low and its unclear why most of the figures are horizontally stretched.

Reply:Thank you very much for your suggestion. It has been revised in the newly submitted paper.

Reviewer 4 Report

Comments and Suggestions for Authors

This paper explored differences in the gut microbiota and associated metabolites between healthy and diarrheic-weaned piglets using macrogenomic and metabolomic analyses. The experimental design and data analysis are generally appropriate. The work is interesting and easy to read. The data are largely supportive of the conclusions. There are only some minor weaknesses.

1.    Please check the full text for writing issues. If p-value appears as, "P-value", "P- values", "p < 0.05", "P < 0.05", etc.

2.    Please check the format of Caudovirales.

3.    Line 275: P-value notation is incorrect, please check and correct it.

4.    The reference number format is incorrect. It is recommended to modify it.

5.    The ruler presented in Figure 1 is somewhat indistinct, thus I propose to replacing it.

6.    Figure 12: the font is so small that I can’t see clearly.

Comments on the Quality of English Language

English language is fine and easy to read.

Author Response

1.Please double check the full text for writing issues. If p-value appears as , "P-value", "P- values", "p < 0.05","P < 0.05", etc.

Reply:Thank you very much for your suggestion. It has been revised in the newly submitted paper.

2.Please double check the full text for writing issues. About the format of Caudovirales.

Reply:Thank you very much for your suggestion. It has been revised in the newly submitted paper.

3.Line 275: P-value notation is incorrect, please check and correct.

Reply:Thank you very much for your suggestion. It has been revised in the newly submitted paper.

4.References: The reference number format is incorrect. It is recommended to modify it.

Reply:Thank you very much for your suggestion. It has been revised in the newly submitted paper.

5.The ruler presented in Figure 1 is somewhat indistinct, thus I propose to replace it.

Reply:Thank you very much for your suggestion. It has been revised in the newly submitted paper.

6.Figure 12: the font is so small that I can’t see clearly.

Reply:Thank you very much for your suggestion. It has been revised in the newly submitted paper.

Reviewer 5 Report

Comments and Suggestions for Authors

As a recommendation, please zoom in each component from figure 2 (Diversity of gut microbiota), to became understandable. Also, the PCA graph can be observed in corroboration with gut composition (figure 3).  

Also, take in consideration the increment of each component from figure 3 for an increased intelligibility, or take in consideration a table with gut composition data.

If it is possible, increase quality of figure 12, for a better observation of marked correlations on the heatmap.

If it possible, increase resolution for all the figures from the paper.

Please, take in consideration reformulation on rows 346-350 (Discussions). Information about villi modification sound somehow redundant, introduced in three phrases.

Author Response

As a suggestion, zoom in on each component in Figure 2 (Diversity of gut microbiota) to make it easy to understand. In addition, the PCA diagram can be observed in corroboration with intestinal components (Figure 3).

Reply:Thank you very much for your suggestion. It has been revised in the newly submitted paper.

Also, take in consideration the increment of each component from figure 3 for an increased intelligibility, or take in consideration a table with gut composition data.

Reply:Thank you very much for your suggestion. It has been revised in the newly submitted paper.

If it is possible, increase quality of figure 12, for a better observation of marked correlations on the heatmap.

Reply:Thank you very much for your suggestion. It has been revised in the newly submitted paper.

If it possible, increase resolution for all the figures from the paper.

Reply:Thank you very much for your suggestion. It has been revised in the newly submitted paper.

Please consider the reformulation of lines 346-350 (discussion). The information about fluff grooming sounds somewhat redundant and is introduced in three phrases.

Reply:Thank you very much for your suggestion. It has been revised in the newly submitted paper.